# Behavioral Effects of the Mixture and the Single Compounds Carbendazim, Fipronil, and Sulfentrazone on Zebrafish (*Danio rerio*) Larvae

**DOI:** 10.3390/biomedicines12061176

**Published:** 2024-05-25

**Authors:** Samara da Silva Gomes, Jadson Freitas da Silva, Renata Meireles Oliveira Padilha, João Victor Alves de Vasconcelos, Luís Gomes de Negreiros Neto, James A. Marrs, Pabyton Gonçalves Cadena

**Affiliations:** 1Department of Morphology and Animal Physiology, Universidade Federal Rural de Pernambuco, Av. Dom Manoel de Medeiros s/n, Dois Irmãos, Recife 52171-900, PE, Brazil; gomesamara27@gmail.com (S.d.S.G.); fs.jadson@gmail.com (J.F.d.S.); renatamopadilha@gmail.com (R.M.O.P.); 2Department of Physics, Universidade Federal Rural de Pernambuco, Av. Dom Manoel de Medeiros s/n, Dois Irmãos, Recife 52171-900, PE, Brazil; jvasconcelosuf@gmail.com (J.V.A.d.V.); lgomes1004@gmail.com (L.G.d.N.N.); 3Department of Biology, Indiana University Purdue University Indianapolis, 723 West Michigan, Indianapolis, IN 46202, USA; jmarrs@iu.edu

**Keywords:** pesticides, systemic effect, behavior, synergism

## Abstract

Pesticides are often detected in freshwater, but their impact on the aquatic environment is commonly studied based on single compounds, underestimating the potential additive effects of these mixtures. Even at low concentrations, pesticides can negatively affect organisms, altering important behaviors that can have repercussions at the population level. This study used a multi-behavioral approach to evaluate the effects of zebrafish larvae exposure to carbendazim (C), fipronil (F), and sulfentrazone (S), individually and mixed. Five behavioral tests, thigmotaxis, touch sensitivity, optomotor response, bouncing ball test, and larval exploratory behavior, were performed to assess potential effects on anxiety, fear, and spatial and social interaction. Significant changes were observed in the performance of larvae exposed to all compounds and their mixtures. Among the single pesticides, exposure to S produced the most behavioral alterations, followed by F and C, respectively. A synergistic effect between the compounds was observed in the C + F group, which showed more behavioral effects than the groups exposed to pesticides individually. The use of behavioral tests to evaluate pesticide mixtures is important to standardize methods and associate behavioral changes with ecologically relevant events, thus creating a more realistic scenario for investigating the potential environmental impacts of these compounds.

## 1. Introduction

Pesticides are a recognized problem that goes beyond human health because they pose serious threats to other vertebrates, invertebrates, and ecosystems in general. The dispersal of these products can reach non-target organisms such as fish, which can suffer physiological changes and, in some cases, can even cause death. In addition, changes in the organism’s behavior, in turn, can reduce the fitness of an individual, leading to population decline and serious effects on the ecosystem [1,2]. The application of these products does not respect natural barriers such as relief and riparian forests, and it is almost impossible to control their course in the field [3].

Recently, a series of studies showed that pesticides including carbendazim, fipronil, and sulfentrazone were detected in aquatic ecosystems and even in food [4,5], indicating that assessing the risks of these pesticides and their mixtures is necessary and important because they are used in common crops [6,7,8]. It is known that the fungicide carbendazim (C_9_H_9_N_3_O_2_) can promote neural excitability (0.57–0.64 mg/L), promoting a hyperactive state in zebrafish larvae, leading to disorganized swimming patterns [9]. Just as the insecticide fipronil (C_12_H_4_C_l2_F_6_N_4_OS) can cause hyperpolarization of neuronal membranes, causing convulsions and muscle spasms (0.50–2.00 mg/L). It can also damage the nervous system, including structures such as the optic thalamus (optic tectum) of aquatic vertebrates [10,11]. The neurotoxic effects of the herbicide sulfentrazone (C_11_H_10_C_l2_F_2_N_4_O_3_S) are not well understood. However, it acts on mitochondrial complex IV and alters cellular energy supply, acting like an endocrine disruptor and affecting zebrafish heart development in the early stages (0.01–0.40 mg/L) [12].

The assessment of these chemicals in relation to the aquatic environment is mainly studied by determining lethal concentrations, which is essential from a regulatory point of view, but reflects only the worst-case scenario [2]. The subtle effects of sublethal exposure are poorly understood, but are important to understand environmental impact [13]. Among sublethal effects, behavior is increasingly being studied. Although the behavioral analysis was initially slow to be integrated into aquatic toxicology, its speed, sensitivity, and environmental relevancy compared to traditional lethality endpoints make it a promising source of information [2,14]. Several studies report behavioral toxic effects at concentrations that are orders of magnitude lower than lethal concentrations of the chemicals [2,15]. Pesticides are commonly found in the aquatic environment at low concentrations, which accumulate, and effects are amplified through the food chain [16].

Zebrafish (*Danio rerio*) is widely used as an animal model for behavioral studies [14,15], not only because of their morphological and genetic conservation with humans (60 to 80% homology) [17,18], but also behavioral similarities [2,19,20], exhibiting a wide range of complex behaviors including social interactions, anxiety, learning, memory, and avoidance behaviors that may be useful for modeling neurological and psychiatric diseases [15].

There is growing interest in the behavior of zebrafish larvae for large-scale testing [21]. A single reproductive cycle can produce hundreds of embryos that rapidly develop, with organs such as the brain, heart, liver, pancreas, kidneys, bones, muscles, and sensory systems maturing within 5 dpf (days post-fertilization) [22]. Behaviors, such as swimming, hunting, and fleeing, are exhibited during the first week of development [23,24]. In addition, neurotoxic agents can be easily studied by exposure to their water during the early developmental period [25].

Despite the obvious difference between zebrafish and humans, the behavioral paradigms of mammals and zebrafish are closely comparable, suggesting an evolutionarily conserved nature in many behaviors across species [19,23]. Even with neuroanatomical differences between mammals and teleost, research shows homologous functions in several areas of the zebrafish brain [26]. Zebrafish larvae exhibit a robust cognitive repertoire, such as memory, social and spatial learning, anxious behaviors, stress, and fear, just as in other vertebrates, and the same neurotransmitters and neuroendocrine systems are present [15,27].

Many tests are used to investigate toxicological effects on behavior, such as basic motor responses, sensorimotor responses, and/or learning and memory [13,28]. However, due to the high complexity of behavioral responses, there is currently no standardized method or assay, which leads to difficulties when comparing behavioral experiments and their results [29]. Therefore, the strategy with multi-behavioral testing is advisable to determine the interactions between nervous system structures [30], since chemicals do not always produce the same behavioral phenotype [31].

Therefore, the present study tests the pesticides carbendazim, fipronil, and sulfentrazone in zebrafish larvae, analyzing their effects through a multi-behavioral analysis, since this can reveal defects not only in neuronal cells, but also in neural functions [21]. Previous studies suggest that these pesticides alone and in combination cause neurotoxic effects in zebrafish [9,10,12,32,33].

## 2. Materials and Methods

### 2.1. Reagents and Solutions

The pesticides used to prepare the test solutions were carbendazim (C) (lot # 002-18-54392, CAS: 10605-21-7, 50% (*w*/*v*)) formulated by Adama Brasil S/A (Londrina, Brazil), fipronil (F) (lot # 001/19, CAS: 120068-37-3, 2.5 (*w*/*w*)) formulated by Rogama Indústria e Comércio LTDA (Pindamonhangaba, Brazil) and sulfentrazone (S) (lot #1041-19-13767, CAS: 122836-35-5, 50% (*w*/*v*)) formulated by FMC Química do Brasil Ltd.a (Paulínia, Brazil), were purchased from commercial suppliers. The three pesticides tested were initially diluted in dimethyl sulfoxide (DMSO) (lot # 85713, CAS: 67-68-5, ≥99.9% purity, Dinâmica Química Contemporânea Ltd.a., Indaiatuba, Brazil), and these solutions were subsequently diluted in embryo medium (8.00 mM NaCl, 0.40 mM KCl, 0.60 mM KH_2_PO_4_, 0.35 mM Na_2_HPO_4_, 0.72 mM CaCl_2_, 1.23 mM MgSO_4_, and 0.35 mM NaHCO_3_ at pH 7.2) [34] to obtain the final concentrations (Table 1). The final nominal concentration of DMSO was less than 0.01% (*v*/*v*), and this is 50 times lower than that reported that affected behavioral parameters in zebrafish [35]. Pesticide concentrations were based on values equal to or lower than the maximum residue limits (MRL) of these pesticides in foods allowed by the Brazilian Health Regulatory Agency—ANVISA [6,7,8]. The pesticide residues were treated by advanced oxidation process (AOP) in a reactor using hydrogen peroxide/ultraviolet irradiation (16 w) before final discard.

### 2.2. Zebrafish Culture and Embryo Production

All protocols involving the animals were approved by the Ethics Committee for the Use of Animals (License No. 7373131021). Adult wild-type fish (1 year) were reared and housed at the Laboratório de Ecofisiologia e Comportamento Animal—LECA vivarium, Universidade Federal Rural de Pernambuco—UFRPE. Zebrafish were quarantined in 80 L aquariums to detect or confirm the absence of pathogens or diseases. They were housed under the following laboratory conditions, artificial aeration of 11 mg/L DO, a temperature of 25 ± 1 °C, pH 7.5 ± 0.5, and a 14/10 h (light/dark) cycle. The water was partially renewed once a week. Abiotic parameters such as dissolved oxygen, ammonia, nitrite, and nitrate were also measured and maintained within ideal ranges [36]. The animals were fed three times a day with 2× Fort Color^®^ fish feed (30% crude protein), and 1× with live brine shrimp nauplii (Artemia ssp). To obtain embryos, zebrafish males and females, in the ratio of 2:1 [34], were placed in the spawning tanks (Alesco^®^ Zebclean, Monte Mor, Brazil) for reproduction. After 30 min from the start of spawning the eggs were collected, the unfertilized eggs were removed, and the fertilized eggs (normal blastula development) [36] were washed and transferred to the exposure chambers.

### 2.3. Chemical Exposure

Toxicity tests were carried out following OECD 210 guidelines [36] with modifications. Briefly, embryos at 2 hpf (hours post-fertilization) were randomly added to sterile polystyrene chambers and exposed to 50 mL of the test solutions (Table 1). Animals were exposed to 144 hpf (or 6 dpf). The solutions were renewed daily, and an embryo medium without pesticides was used as a control. We chose not to feed the larvae, as they live up to 7 dpf exclusively on the nutrients provided by the yolk. According to Clift et al. [37], larvae fed at 6 and 7 dpf showed changes in swimming and resting speed compared to their non-fed congeners.

### 2.4. Behavioral Tests

Behavioral tests were conducted between 11 am–12 noon [38] using a filming system equipped with a camera (Canon 6D Mark and lens EF 75–300 mn f/4–5.6 III), LED board, and/or LCD screen to playback and record the videos of larvae. In total, 15 larvae were used for each experimental group, and all larvae were tested simultaneously and randomly and went through the same sequence of behavioral tests: thigmotaxis, touch sensitivity, optomotor response, exploratory behavior of larvae, and bouncing balls, respectively. Each test was repeated seven times, one replicate per week (*n* = 105 per group with 8 experimental groups ≈ 840 larvae). Behavioral tests were conducted only with larvae without visible morphological teratogenic effects [39,40]. Based on the previous screening test, pesticide concentrations used in this study did not produce mortality or significant morphological teratogenic effects. Initially, the larvae were distributed in 48-well plates, and subsequently the same larvae were transferred to 6-well plates, where they were organized into groups of 5 larvae per well for conducting a bouncing ball test. The behavioral evaluation of the larvae was performed through subsequent analysis of the videos recorded during the tests described below and performed in the sequence on the same day, respectively (Figure 1).

#### 2.4.1. Thigmotaxis

The thigmotaxis test was used to analyze the propensity of the larvae to approach the walls of the plates to evaluate behavior similar to anxiety in zebrafish [23,39] because when the larva enters an unfamiliar environment, they tend to swim against the walls of the well to explore the new environment [40]. Before behavioral tests, the larvae underwent 15 min of acclimatization in 48-well plates in an environment with natural lighting, and each well was filled with 1 mL of the same test solution to which the animals were previously exposed. The response to the thigmotaxis test was recorded by photography so that all larvae could be evaluated in the same time interval. This response was measured according to the location of the larvae in the well, whether they were at the edge (positive) or in the center of the well (negative).

#### 2.4.2. Touch Sensitivity

The touch sensitivity test was used to evaluate the larval response to mechanical stimuli. The larvae were gently touched in the posterior portion of the tail, and the response was recorded if the larva exhibited escape behavior. An escape behavior response was determined when larvae displayed swimming behaviors (positive) [39], and no response when larvae remained motionless after the touching (negative).

#### 2.4.3. Exploratory Behavior of Larvae

The exploratory behavior of the larvae was evaluated during spontaneous swimming (without stimuli), where the larvae tend to swim at the limits of the wells (close to the wall), a tendency that is explained by the initial exploratory behavior and the avoidance of the central area of the well, where there is less protection (thigmotaxis). This assay was based on the protocol by Altenhofen et al. [41] and Pérez-Escudero et al. [42], with modifications. After 15 min of acclimatization, the exploratory activity of the larvae was observed in the 48-well plates for 5 min with artificial lighting (LED plate) without stimuli. However, the first and last minute of recording were excluded from the analysis, as it was observed that in the first minute, the larvae did not show swimming behaviors due to exposure to artificial light from the LED board, and in the last minute, the animals reduced their display of swimming behaviors, as was also observed by [42]. During the analysis, the animals that remained motionless during the recording were counted, and those that showed some swimming movement had their average speed (cm/s) and total trajectory covered (cm), as measured using the IdTracker software v 8.3 [43]. These were the parameters considered for exploring the new environment in this assay.

#### 2.4.4. Optomotor Response

The optomotor response is used to measure the innate visual responses of larvae in which the individual swims in the direction of the optical flow and mimics naturalistic behavior, aiming to stabilize itself according to the flow of the video lines. The methodology was based on Cadena et al. [39] and Brastrom et al. [44], with modifications. The plates with animals were placed on a 21.5 inch LCD screen (Dell E2211H) in a room without external sounds and lights. The fish were acclimated for 15 min on a dark screen, and then the animation was started (video as black and white lines moving from right to left for 30 s and then left to right for another 30 s). Subsequently, the video recording was analyzed, and larval positioning was classified into two types with respect to the flow of the lines: low alignment (70% of the well), and high alignment (30% of the well), according to Cadena et al. [39].

#### 2.4.5. Bouncing Ball Test

The bouncing ball test is used to observe the spatial interaction and social and escape responses of larvae to visual stimuli [21,45,46]. Larvae were transferred to 6-well plates (five larvae per well) and placed on an LCD screen (Dell E2211H), as described above. An animation with visual stimuli created in Microsoft PowerPoint v 2404 (Office 365) was played. The animation had red balls (diameter 1.35 cm) that moved from left to right on a square with a 2 cm trajectory in the bottom half of the well area (stimulus area) for 5 min after a 15 min acclimation on a dark screen. The percentage of larvae in the non-stimulated area (upper part of the well) during the 5 min session was taken to be indicative of the cognitive ability of the group to exhibit escape behavior against the red ball. The number of larvae in the same quadrant of the well was used to measure the percentage of grouping (positive, if three or more larvae were grouped) and social interaction among the larvae in the well (positive, if larvae show escape behavior from red ball). A time-point sampling method was used to collect the data (one analysis every 30 s) [45,46].

### 2.5. Statistical Analysis

Statistical analyses were performed using Origin Pro Academic 2015 (Origin Lab. Northampton, MA, USA). The normality was determined with the Shapiro–Wilk test with *p* < 0.05, and Levene’s test for homogeneity of variance was carried out on each variable. All data were presented as mean ± SD. For behavioral tests of thigmotaxis, sensitivity to touch, exploratory behavior, and bouncing balls, we used pooled samples from several animals. A one-way ANOVA was used for data that presented a normal distribution, followed by the Tukey test with *p* < 0.05 to determine the significant difference between groups. To evaluate the interactions of pesticides in the trajectory followed during the exploratory activity test, the simplex centroid design method was carried out using Statistica 14 (TIBCO, Palo Alto, CA, USA) according to Cadena et al. [40]. The results of the optomotor response test for each animal were analyzed using Bowker’s non-parametric test with *p* < 0.05 [44].

### 2.6. Cluster Analysis

The tests were evaluated using one-way ANOVA for the seven behaviors observed. For the results obtained, a number between 1 and 3 was assigned based on their statistical *p*-values, according to Richendrfer and Creton [47], with some modifications. Any behavior that was statistically different from the control was given the numbers 1 or 2 (1 for *p* < 0.01 or 2 for *p* < 0.05). Behaviors that were not statistically different from the control were given the number 3 (*p* > 0.05). The numbers assigned to the behavior scores were imported into Origin Pro Academic 2015 software (Origin Lab. Northampton, MA, USA). Colors were assigned to the numbers to illustrate; the green color corresponds to numbers 1 and 2 (*p* < 0.01 and *p* < 0.05) and black corresponds to 3 (*p* > 0.05). The results of the Optomotor Activity Test were not included in this analysis because the data were analyzed using a non-parametric test (Bowker’s test), and thus there was no *p*-value against the control to be compared.

## 3. Results

### 3.1. Thigmotaxis and Touch Sensitivity Tests

Larvae that were exposed to single pesticides and their mixtures. Thigmotaxis (Figure 2a) and touch sensitivity (Figure 2b) tests showed no significant difference from the control group.

### 3.2. Exploratory Behavior of Larvae

In this test, first, the animals that remained immobile (Figure 3a) during all the time intervals of the test were separated and used to determine the percentage of the immobile animals. Group F was the one that showed a significant difference, with a higher percentage of immobile animals. Later, with the animals that showed some swimming movement, the average speed (Figure 3b) and the total distance traveled (Figure 3c) were evaluated. Groups S, C + F, and F + S showed a significant difference in mean velocity, and these same groups showed a significant difference in the total distance traveled compared to the control. Results showed that groups with the lowest velocity (Figure 3b) were also groups of larvae with the shortest trajectory (Figure 3c). It was also observed that the groups C + F and F + S showed a synergistic effect (Figure 3d), as they showed a reduction compared to groups exposed to C and F individually. It was also possible to qualitatively evaluate the trajectory (Figure 4) of the larvae in each group, and we saw that even the groups that did not show a significant reduction compared to the control in their trajectory showed an abnormal trajectory during the tests.

### 3.3. Optomotor Response Test

All larvae exhibit optomotor responses in the first period of analysis with right-to-left line movement (Figure 5). However, it was observed in the optomotor response test that the larvae exposed to the pesticides F and S alone, the mixtures C + F and C + S showed no significant difference in the second period of analysis with left to right line movement compared to the control. The results demonstrated that these larvae exhibited low alignment with the direction of movement of the stripes and did not respond normally to the animation, indicating that these pesticides caused a deficiency in the optomotor system of zebrafish larvae. Interestingly, it was also observed that the groups exposed to the mixtures C + F and C + S showed no significant difference, suggesting that when the pesticides F and S are mixed with C, they have a synergistic effect, resulting in a higher percentage of larvae with low alignment compared to when the larvae were exposed to the three pesticides individually. However, when F and S were mixed (F + S), an antagonistic effect was observed, reducing the percentage of larvae with low alignment compared to that observed with the two single pesticides.

### 3.4. Bouncing Balls Test

In the bouncing balls test, two behaviors were observed: escape from the visual stimulus (Figure 6a) and the grouping (Figure 6b) of the larvae. All exposed groups showed a significant difference compared to the control in the two analyzed behaviors. About escape from the visual stimulus, larvae exposed to the single pesticides and their mixtures stayed significantly longer in the lower part of the well (area with stimulus) and did not show evasive behavior relative to the red ball, as the control did.

### 3.5. Cluster Analysis

The analysis of the behavioral results clustering by exposure group is described in Figure 7. Cluster analysis identifies the groups that presented the greatest number of different behaviors from the control. Based on this, we observed that among the isolated compounds, sulfentrazone was the group that presented the most differences from the control, followed by fipronil, and then carbendazim. We also saw that the binary mixture C + F showed more differences than the isolated compounds, even though it was 50% of the isolated concentration, suggesting that these mixed compounds potentiated the toxic effects in zebrafish larvae. This was similar to the group that was exposed to the binary mixture of F + S.

## 4. Discussion

Our study used a multi-behavioral approach to assess how single and mixed pesticides affected zebrafish larvae. Interestingly, this approach was very sensitive as a tool, and we recommend its inclusion in the toxicological tests. In addition, the analysis of different behavioral parameters can help to estimate more accurately the impact of pesticides on zebrafish behavior. The changes in the behavior result from one or a combination of molecular, biochemical, and physiological changes [48].

The group exposed to sulfentrazone was the one that showed the highest number of behavioral changes (Figure 7). Showed significant differences in mean speed (Figure 3b), distance traveled (Figure 3c), and presented an abnormal trajectory (Figure 4). According to Jiang et al. [12], exposure to sulfentrazone (0.01–3.60 mg/L for 30 days) produced adverse effects on calcium (Ca^2+^) channel regulation in zebrafish larvae, affecting the activities of Ca^2+^-ATPase, total Na^+^K^+^-ATPase, and Ca^2+^Mg^2+^-ATPase, as well as induced the transcription of many key calcium manipulation proteins, such as ATPase, Ca^2+^, transporter subunits Atp2b2, Atp2b3a, and Atp2b3b, and ATPase Na^+^/K^+^ transporter subunit Atp1b4. This biochemical process participates in muscle mechanics, and its alteration can compromise the locomotor performance of zebrafish [48]. Because the interaction of actin and myosin depends on the promotion of Ca^2+^, where Ca^2+^ is exported with high specificity by Ca^2+^-ATPases in the plasma membrane and the deregulation of Ca^2+^-ATPase activity, it can cause defects in the locomotor system of zebrafish [49]. We suggest that the alterations described above induced by sulfentrazone in zebrafish may contribute to the locomotor deficit observed in our study. Changes in swimming behavior can have important consequences for feeding behaviors (prey capture) and susceptibility to predation (predator escape), among others [48].

Larvae exposed to fipronil showed a significant increase in immobile behavior during the exploratory activity test (Figure 3a) and showed an abnormal trajectory in the test (Figure 4). Immobility of zebrafish was also observed by Wu et al. [10] when analyzing adult zebrafish exposed to fipronil (0.50–2.00 mg/L). They saw that after 24 h of exposure, there was a significant reduction in swimming and the traveled distance by the animals. The authors attributed these locomotion abnormalities to the neurotoxicity of fipronil in zebrafish brain tissue, possibly due to oxidative neural stress, inflammation, and apoptosis of neurons. This was confirmed with analysis of markers of oxidative stress (SOD2), inflammation (TNF-α), and apoptosis (caspase-3), showing a significant decrease in SOD2 and TNF-α levels and a significant increase in caspase-3 in zebrafish brain tissue. Park et al. [32] observed that fipronil caused developmental delays and motor neurons.

Fish generally adopt self-defense behaviors, like remaining motionless near the bottom or walls [50], which is an effective strategy to avoid capture by predators. However, fipronil resulted in the zebrafish larvae losing their ability to move, making larvae more vulnerable to predators. This group also showed significant differences in the optomotor response tests (Figure 5) and bouncing balls (Figure 6). Fipronil acts directly on γ-aminobutyric acid, preventing the closure of chloride channels (GABA) [51]. Reduced GABA receptor signaling may lead to loss of retinal ganglion cells [52]. Fipronil may impair the visual system.

Among the groups exposed to a single pesticide, carbendazim was the least toxic (Figure 7). The group exposed to carbendazim did not show reductions in average speed and traveled distance, but an abnormal trajectory was observed in the exploratory activity test (Figure 4). This change in swimming pattern was also observed by Zhang et al. [9]. The researchers associated changes in swimming behavior with changes in gene expression, namely the ctsbl gene, which plays a crucial role in antigen processing, regulation of cell death (apoptosis), autophagy, and metabolism, contributing to increased cell death mainly in the nervous system [9].

The carbendazim group also showed significant differences in the optomotor response test (Figure 5) and bouncing ball (Figure 6a,b). This result may be related to the impairment of the visual system in this exposed group. *Xenopus laevis* (clawed frog) larvae exposed to carbendazim when analyzed histologically by Yoon et al. [53] showed dysplasia and optic edema as the most common malformations. Furthermore, it was also observed that the retinal layers were poorly differentiated and thinner and the neural cells in the larval brain were disaggregated and invaded the cerebral ventricle, demonstrating that carbendazim strongly inhibited the differentiation of neural tissue. The authors suggest that this inhibition may be related to the interruption of tubulin synthesis, weakening cell junctions. Carbendazim interferes with the assembly of tubulin and the formation of microtubules, resulting in various malformations, genetic toxicity, and reproductive problems [54].

Our results showed that larvae exposed to binary mixtures C + F and F + S exhibited synergistic responses in zebrafish (Figure 3d) in the behavioral parameters affected in our work. Although the concentration of the binary mixture was 50% lower than the single pesticide groups, we observed reduced speed (Figure 3b), distance traveled (Figure 3c), and optomotor response (Figure 5). Synergistic toxicity is often found in organisms exposed to pesticide mixtures [55]. Larvae exposed to mixtures may experience oxidative stress, leading to the production of reactive oxygen species (ROS) [56], which could help explain our results. Several studies of pesticide mixtures with different modes of action have pointed to increased ROS production in zebrafish because of the synergy between pesticides [55,56,57]. Excessive ROS production was associated with apoptotic cell death in zebrafish embryos and larvae [58].

The binary and ternary mixtures, C + S, F + S, and C + F + S, also caused significant behavioral changes. We observed different toxicity patterns for mixtures depending on the parameters analyzed, where synergistic effects were observed. This type of response was also observed in other studies, where the effects on the same organism varied depending on the chosen endpoint [59,60]. In this way, we can also see that behavioral tests are useful tools and have the potential to reveal environmental stress and how combinations of effects produce heterogeneous behavioral changes in the environment. Future studies could elucidate the additive and synergistic mechanisms produced by mixed pesticides.

## 5. Conclusions

This study addressed the impact of pesticides, both individually and in mixtures, on zebrafish larvae using a multi-behavioral approach. This approach proved to be a sensitive and comprehensive tool for analyzing the systemic effects of one or more substances in zebrafish larvae. The results showed that among the individual pesticides, sulfentrazone caused the greatest number of behavioral changes in exposed larvae. Fipronil, on the other hand, caused immobile behavior. Carbendazim had less pronounced effects, but still affected the animals’ escape and grouping behavior. When we analyzed pesticide mixtures, we observed synergistic responses that affected speed, distance covered, and optomotor response. These results highlight the complexity of pesticide mixture toxicity and the sensitivity of behavioral tests, which can be used as initial indicators of environmental stress. Experiments testing pesticide mixtures will help us understand their additive and synergistic environmental exposure effects.

## Figures and Tables

**Figure 1 biomedicines-12-01176-f001:**
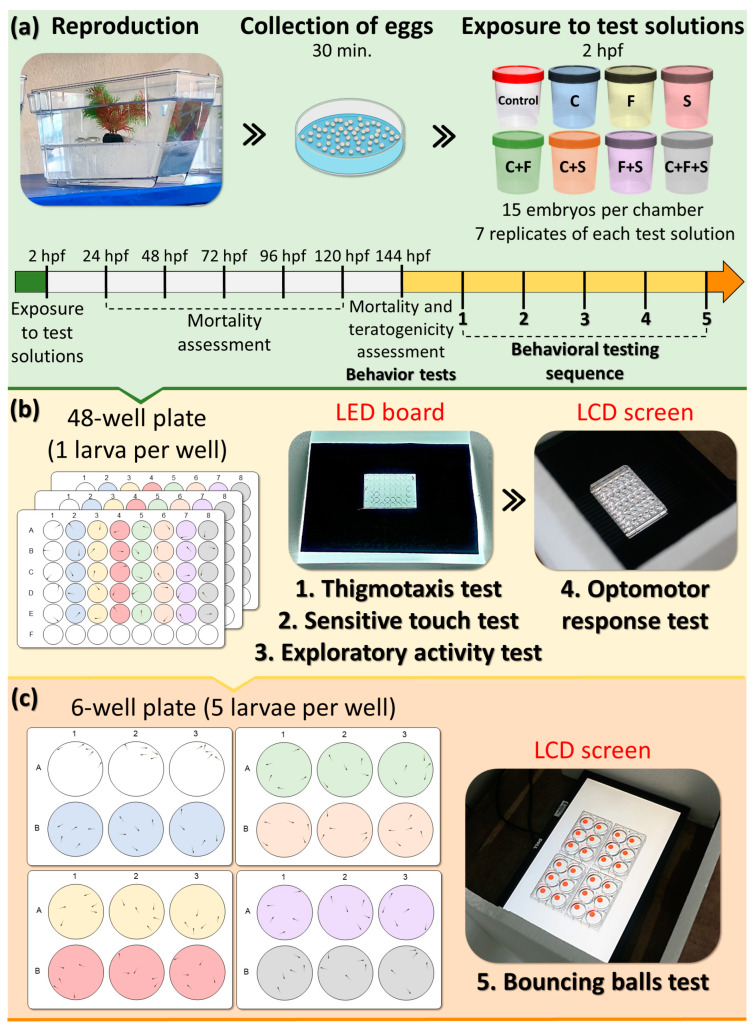
Experimental design. (**a**) After breeding and collection, the embryos were randomly allocated into experimental groups, with 15 animals per test solution. (**b**) After 144 h post-fertilization (hpf), larvae were randomly removed from polystyrene containers and placed in 48-well plates—one larva per well, with five larvae from each group per plate. After a period of acclimatization, the plates (one at a time) were positioned on an LED plate for video recording and photography. These videos and photographs were later analyzed for thigmotaxis, sensitivity to touch, and exploratory behavior. The LED board was replaced with an LCD screen, and after the acclimatization period, each of the boards was positioned on the screen to reproduce the animation of the optomotor activity test and record the videos for later analysis. (**c**) Larvae from each group were removed from the wells of 48-well plates and transferred to 6-well plates, with five larvae per well. After the acclimation period, the 6-well plates were placed on top of the LCD screen to play the animation of the bouncing ball test and record videos for later analysis. During all tests, the animals remained in the same test solution. The colors of the wells correspond to the test solutions to which the larvae are exposed.

**Figure 2 biomedicines-12-01176-f002:**
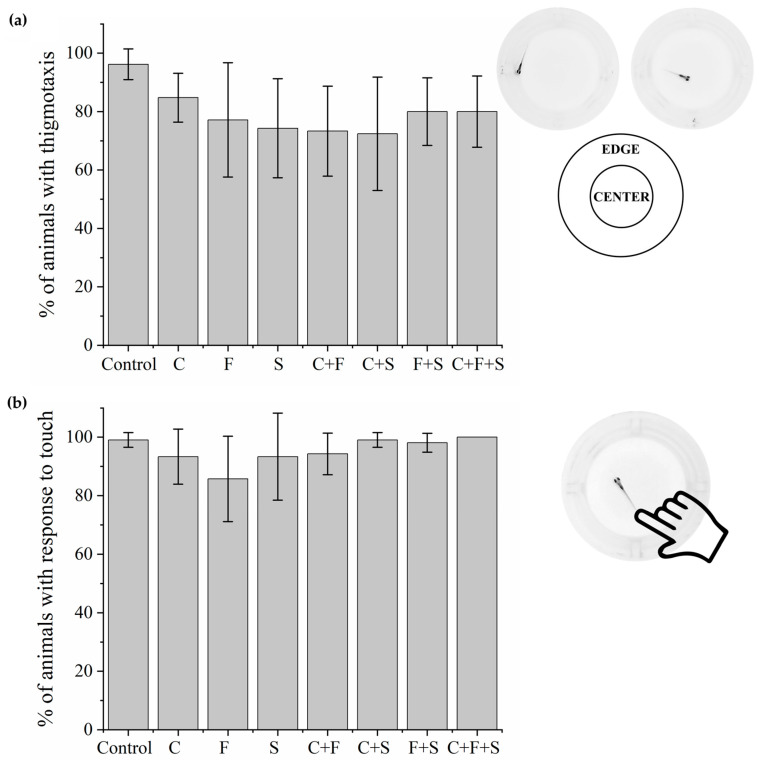
Mean total percentage and standard deviation (SD) of responses of larvae exposed to pesticides and their mixtures in the thigmotaxis test (**a**) and responses of each group compared to the control group using a one-way ANOVA (F (7,55) = 2.06 *p* < 0.07) followed by Tukey’s test (C *p* = 0.82; F *p* = 0.24; S *p* = 0.11; C + F *p* = 0.08; C + S *p* = 0.06; F + S *p* = 0.43; C + F + S *p* = 0.43). Touch sensitivity response test (**b**) and responses of each group compared to the control group using a one-way ANOVA (F (7,55) = 2.11863 *p* < 0.05935) followed by Tukey’s test (C *p* = 0.92; F *p* = 0.09; S *p* = 0.92; C + F *p* = 0.97; C + S *p* = 1; F + S *p* = 1; C + F + S *p* = 1). Legend: Contr—Control; C—Carbendazim; F—Fipronil; S—Sulfentrazone.

**Figure 3 biomedicines-12-01176-f003:**
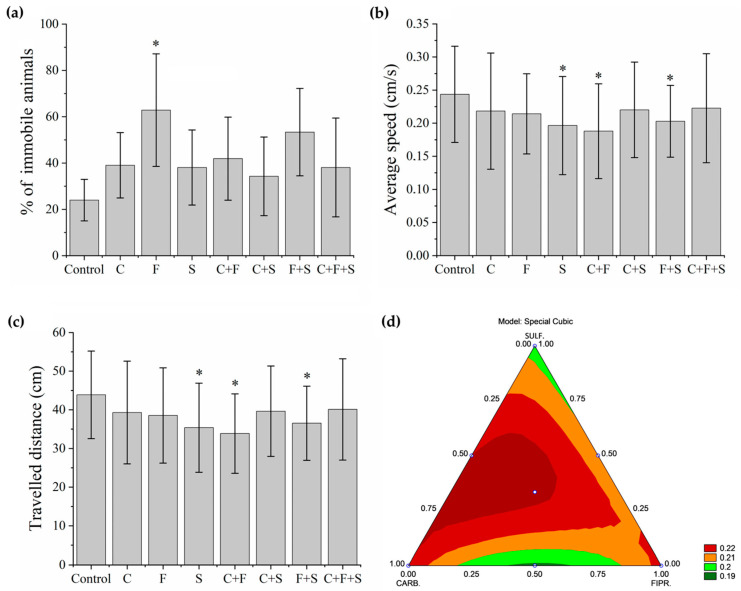
Mean total percentage and SD of larvae that remained immobile during the Exploratory Activity test (**a**). Groups were compared to the control group using a one-way ANOVA (F (7,55) = 2. 71277, *p* < 0.01) followed by Tukey’s test (C *p* = 0.84; F *p* = 0.01; S *p* = 0.88; C + F *p* = 0.70; C + S *p* = 0.98; F + S *p* = 0.13; C + F + S *p* = 0.88), *p* < 0.05 (*). Mean Speed of larvae swimming during the test (**b**). Groups were compared to the control group using a one-way ANOVA (F (7,435) = 4.0053, *p* < 0.01) followed by Tukey’s test (C *p* = 0.44; F *p* = 0.37; S *p* = 0.002; C + F *p* < 0.05; C + S *p* = 0.52; F + S *p* = 0.03; C + F + S *p* = 0.67), *p* < 0.05 (*). Mean Total Distance traveled by larvae that swam during the test (**c**). Groups were compared to the control group using a one-way ANOVA (F (7,435) = 4. 00565, *p* < 0.01) followed by Tukey’s test (C *p* = 0.44; F *p* = 0.37; S *p* = 0.003; C + F *p* < 0.05; C + S *p* = 0.52; F + S *p* = 0.03; C + F + S *p* = 0.68), *p* < 0.05 (*). (**d**) Ternary contour plot of the variables Carbendazim, Fipronil, and Sulfentrazone for a special cubic model (F (3,380) = 3.004, *p* = 0.03), evaluating the individual effects and interactions of these pesticides in their mixtures. Carbendazim and Sulfentrazone were more toxic by reducing the total distance traveled (green areas). Key: Contr—Control; C—Carbendazim; F—Fipronil; S—Sulfentrazone.

**Figure 4 biomedicines-12-01176-f004:**
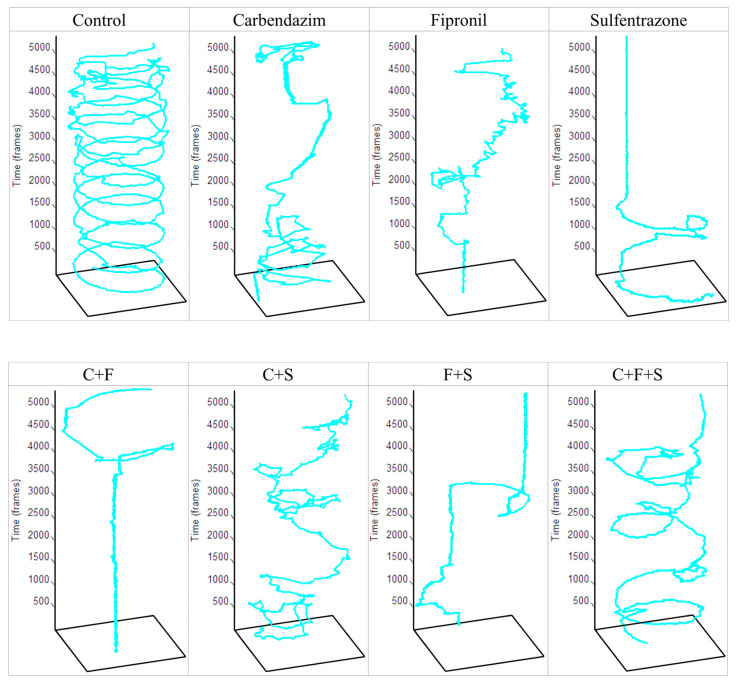
Total trajectory traveled by a zebrafish larva from each experimental and control group in one well of a 48-well plate during the 3 min recording of the Exploratory Activity test. Key: Contr—Control; C—Carbendazim; F—Fipronil; S—Sulfentrazone.

**Figure 5 biomedicines-12-01176-f005:**
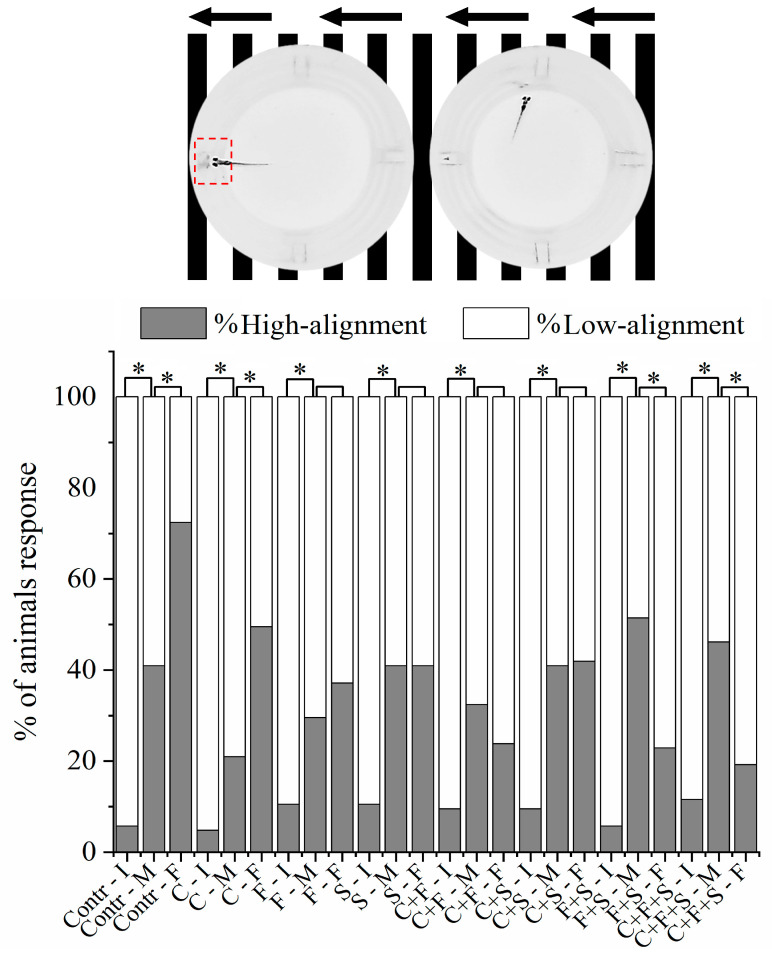
Mean total percentage of zebrafish larvae alignment to the well during the Optomotor Response test. If the larva is inside the red demarcated area, it is considered high alignment, and outside the area is low alignment. It was analyzed by Bowker’s symmetry test, *p* < 0.05 (*), to the same group among initial, middle, and final test positions. (Contr *p* < 0.05; C *p* < 0.05; F *p* = 0.16; S *p* = 1; C + F *p* = 0.16; C + S *p* = 0.88; F + S *p* < 0.05; C + F + S *p* < 0.05). Key: Contr—Control; C—Carbendazim; F—Fipronil; S—Sulfentrazone; I—Initial; M—Middle; F—Final.

**Figure 6 biomedicines-12-01176-f006:**
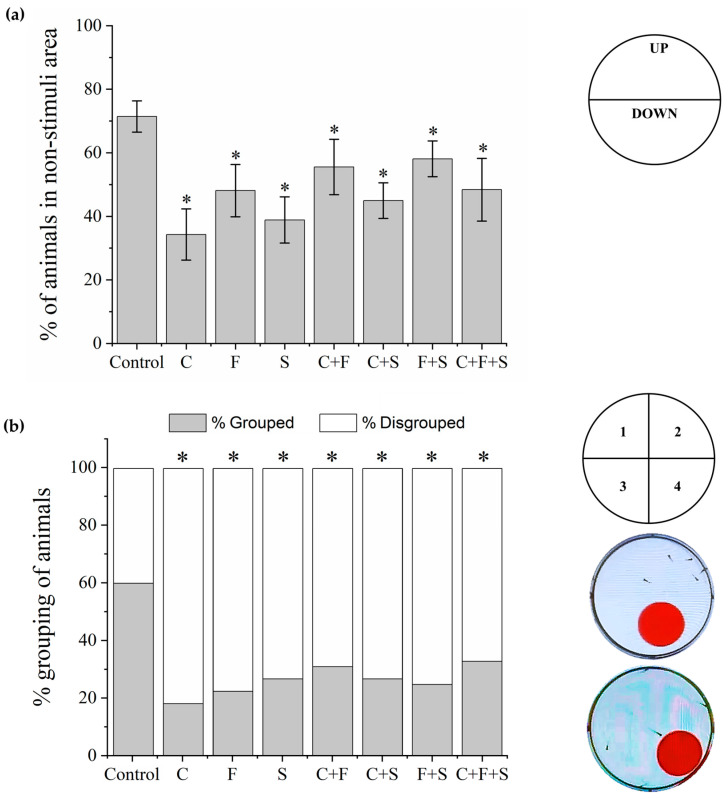
Mean total percentage and SD of larvae that exhibited escape behavior about the visual stimulus and stayed longer in the upper part of the well, area without stimulus, during the Bouncing Ball test (**a**). The response of each group was compared to the control group by one-way ANOVA (F (7,55) = 17.11, *p* < 0.01) followed by Tukey’s test (C *p* < 0.05; F *p* < 0.05; S *p* < 0.05; C + F *p* = 0; C + S *p* < 0.05; F + S *p* = 0.03; C + F + S *p* < 0.05), *p* < 0.05 (*). Percent grouping of larvae during the Bouncing Ball test (**b**), groups were compared to the control group using a one-way ANOVA (F (7,55) = 6.20, *p* < 0.01) followed by Tukey’s test (C *p* < 0.05; F *p* < 0.05; S *p* < 0.05; C + F *p* < 0.05; C + S *p* < 0.05; F + S *p* < 0.05; C + F + S *p* = 0.01), *p* < 0.05 (*). Key: C—Carbendazim; F—Fipronil; S—Sulfentrazone.

**Figure 7 biomedicines-12-01176-f007:**
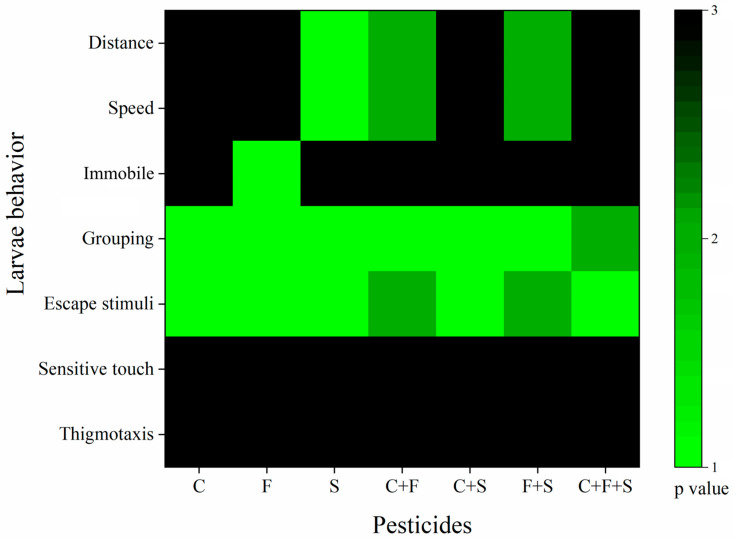
Cluster analysis of behaviors of larvae exposed to single and mixed pesticides at 6 dpf. *p* value compared to the control group as a variable. The green bars indicate values of 1–2, and the black bars indicate a value of 3. Key: C—Carbendazim; F—Fipronil; S—Sulfentrazone.

**Table 1 biomedicines-12-01176-t001:** Concentrations of the single compounds, binary, and ternary mixtures used to measure behavioral changes in zebrafish larvae exposed to the pesticides carbendazim (C), fipronil (F), and sulfentrazone (S). The layout of the simplex centroid design method for evaluating the interactions between the chemical compounds.

Simplex Centroid Design for Behavioral Assessment in 6 dpf (Days Post-Fertilization)			
Components	Carbendazim (mg/L)	Fipronil (mg/L)	Sulfentrazone (mg/L)
C	0.200	0.0	0.0
F	0.0	0.050	0.0
S	0.0	0.0	0.200
C + F	0.100	0.025	0.0
C + S	0.100	0.0	0.100
F + S	0.0	0.025	0.100
C + F + S	0.060	0.017	0.025

## Data Availability

The data and material used in this study are available from the corresponding author on request. The data are not publicly available due to complexity and lack of a standard format.

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
