# Peer review of "Behavioral Effects of the Mixture and the Single Compounds Carbendazim, Fipronil, and Sulfentrazone on Zebrafish (Danio rerio) Larvae"

_biomedicines, 2024, doi:10.3390/biomedicines12061176_

Round 1

Reviewer 1 Report (Previous Reviewer 1)

Comments and Suggestions for Authors

Dear authors, your revisions have mostly addressed the concerns raised, and I am pretty satisfied with the changes made. The article should be accepted in the present form

Author Response

Dear authors, your revisions have mostly addressed the concerns raised, and I am pretty satisfied with the changes made. The article should be accepted in the present form.

Thank you very much.

Reviewer 2 Report (Previous Reviewer 2)

Comments and Suggestions for Authors

The manuscript by Samara da Silva Gomes and colleagues explore the Behavioral Effects of the Mixture and the Single Compounds Carbendazim, Fipronil, and Sulfentrazone on Zebrafish (Danio rerio) Larvae.The role of pesticides as emerging environmental contaminants of emerging concern is a very interesting topic especially when it comes to cocktails of substances at very low concentrations but that in combination can alter the functionality of an organism exposed to them. I understand that this study focuses on the behavioral aspect, nevertheless it would be appropriate to add at least a mortality rate and a hatching rate. The authors spotted mortality during the exsperiment? delay in hatching? some sort of deformity? I strongly believe that a graph or a table are required to assess these data. Finally, a comparison between control and exposed groups is suggested (maybe with some images)

Author Response

The manuscript by Samara da Silva Gomes and colleagues explore the Behavioral Effects of the Mixture and the Single Compounds Carbendazim, Fipronil, and Sulfentrazone on Zebrafish (Danio rerio) Larvae. The role of pesticides as emerging environmental contaminants of emerging concern is a very interesting topic especially when it comes to cocktails of substances at very low concentrations but that in combination can alter the functionality of an organism exposed to them. I understand that this study focuses on the behavioral aspect, nevertheless it would be appropriate to add at least a mortality rate and a hatching rate. The authors spotted mortality during the experiment? delay in hatching? some sort of deformity? I strongly believe that a graph or a table are required to assess these data. Finally, a comparison between control and exposed groups is suggested (maybe with some images).

We studied the mortality and several sublethal effects. We wrote a paper and submitted to Ambiente & Água - An Interdisciplinary Journal of Applied Science ISSN 1980-993X. At this moment this paper is under review. Based on these results, we choose the pesticides concentrations for the current paper. The paper submitted to Biomedicines is focused on behavioral endpoints using only larvae without visible morphological teratogenic effects. We add the information: "Behavioral tests were conducted only with larvae without visible morphological teratogenic effects [39,40]. Based on the previous screening test, pesticide concentrations used in this study did not produce mortality or significant morphological teratogenic effects."

Thank you very much.

This manuscript is a resubmission of an earlier submission. The following is a list of the peer review reports and author responses from that submission.

Round 1

Reviewer 1 Report

Comments and Suggestions for Authors

The manuscript presents interesting findings regarding the effects of pesticide exposure. However, several key points require clarification and additional experimental details

L.46-52. Please indicate the concentrations of pesticides that induced such effects. In general,  it  is  crucial to indicate the concentrations of pesticides that induced the observed effects. This information is essential for a comprehensive understanding of the study's outcomes and allows readers to evaluate the relevance of the findings. Hovewer, there is a lack of such information in the discusson part particularly.
The decision to set dissolved oxygen (DO) at 11 mg/L needs justification, especially considering the natural DO levels in freshwater. In freshwater  DO  reaches approximately 9.1 and 8.3 mg/L at 20 and 25°C, respectively, and 1 atm pressure. At temperatures of 20 and 30 °C, the level of saturated DO is 9.0-7.0 mg/L.
The attempt to explain the behavioral toxicity of fipronil based on other findings without specifying concentrations raises concerns. You also tried to equiped you discussion with previous findings regarding biochemical measurements in adults on larvae. It needs to be figured out
The discussion is rich in speculation about potential visual complications, neural disturbances, and reactive oxygen species (ROS) overflow. However, there is a notable lack of direct experimental evidence supporting these hypotheses. Additional measurements and experimental data are desired to strengthen the proposed explanations.

Comments on the Quality of English Language

The minor revision is needed

Author Response

We are grateful for your careful review and critique of the manuscript. Your concerns are addressed below.

46-52. Please indicate the concentrations of pesticides that induced such effects. In general, it is crucial to indicate the concentrations of pesticides that induced the observed effects. This information is essential for a comprehensive understanding of the study's outcomes and allows readers to evaluate the relevance of the findings. However, there is a lack of such information in the discussion part particularly.

Response: We added pesticide concentrations to the paragraph according to your suggestions. Thanks.

The decision to set dissolved oxygen (DO) at 11 mg/L needs justification, especially considering the natural DO levels in freshwater. In freshwater DO reaches approximately 9.1 and 8.3 mg/L at 20 and 25°C, respectively, and 1 atm pressure. At temperatures of 20 and 30 °C, the level of saturated DO is 9.0-7.0 mg/L.

Response: We use 80-liter aquariums with a density of 1 fish per liter, including a pump to inject O2 into the aquarium, producing greater oxygen availability, which increases comfort and reduces stress for the fish. We obtain values of 15 mg/L in freshwater.

Reference:

https://www.sciencedirect.com/topics/earth-and-planetary-sciences/dissolved-oxygen

https://www.fondriest.com/environmental-measurements/parameters/water-quality/dissolved-oxygen/

The attempt to explain the behavioral toxicity of fipronil based on other findings without specifying concentrations raises concerns. You also tried to equiped you discussion with previous findings regarding biochemical measurements in adults on larvae. It needs to be figured out The discussion is rich in speculation about potential visual complications, neural disturbances, and reactive oxygen species (ROS) overflow. However, there is a notable lack of direct experimental evidence supporting these hypotheses. Additional measurements and experimental data are desired to strengthen the proposed explanations.

Response: We specified the concentrations of fipronil that were missing in the articles cited in the discussion, thank you for the suggestion!

The central idea of our article was to use behavioral tests to analyze effects of the compounds. However, we had difficulty finding articles with methodology or compounds similar to our study.  Several studies indicate that the brain of organisms during an early development are more sensitive to chemicals (dAmora and Giordani, 2018) and that zebrafish larvae are as sensitive as adults (Rosa et al. 2022). Therefore, we think that it is valid to make the comparison between larvae and adults. For example, Wu et al. (2021) reported immobility behavior in adults, just as we observed in larvae.

We found that this multi-behavioral analysis is promising. However, future biochemical experiments will be carried out. We emphasize that using our methodology, systemic effects of fipronil, other pesticides and their mixtures could be detected.

Reference:

d'Amora, M.; Giordani, S. The Utility of Zebrafish as a Model for Screening Developmental Neurotoxicity. Front Neurosci 2018, 12, 976, doi:10.3389/fnins.2018.00976

Rosa, J.G.S.; Lima, C.; Lopes-Ferreira, M. Zebrafish Larvae Behavior Models as a Tool for Drug Screenings and Pre-Clinical Trials: A Review. Int J Mol Sci 2022, 23, doi:10.3390/ijms23126647

Wu, C.H.; Lu, C.W.; Hsu, T.H.; Wu, W.J.; Wang, S.E. Neurotoxicity of fipronil affects sensory and motor systems in zebrafish. Pestic Biochem Physiol 2021, 177, 104896, doi:10.1016/j.pestbp.2021.104896.

Reviewer 2 Report

Comments and Suggestions for Authors

The article : Behavioral Effects of the Mixture and the Single Compounds Carbendazim, Fipronil, and Sulfentrazone on Zebrafish (Danio rerio) Larvae, focuses on behavioral aspects of zebrafish larvae in response to exposure to environmental contaminants. It is certainly of some relevance to study interactions between environmental contaminants (which are increasingly ubiquitous) on animal models even very similar (genetically) to humans such as the teleost Danio rerio. However, I have some concerns :

-The authors have to add the embryo medium composition

-The address of all the manufacturers have to be added

-The organization of the experimental groups, the number of specimens used for each test and the subdivisions of replicates should be clearer and divided according to the tests as well.In order to improve the quality of the manuscript i suggest the authors to take a cue from this study and cite it.https://doi.org/10.3390/toxins14080518

-My biggest concern is for the figure 3. how come in all the graphs the standard deviation is so high? is there an explanation for this result? 

Moreover,I have doubts about these graphs, group F for example has the highest % of motionless animals compared to the control, but they also have high speed and good distance traveled if we compare them with other groups, please explain.

Still in the figure 3 (a) how  the F+S group is not significant compared to the control? from the graph it seems quite different.Please explain this data

Author Response

Reviewer #2:
We are grateful for your careful review and critique of the manuscript. Your concerns
are addressed below.
1. The authors have to add the embryo medium composition
R. We added the composition (L. 109 – 111) to the text according to your suggestion.
Thanks.
2. The address of all the manufacturers have to be added
R. The information was added to the text. Thanks!
3. The organization of the experimental groups, the number of specimens used for each
test and the subdivisions of replicates should be clearer and divided according to the
tests as well.In order to improve the quality of the manuscript i suggest the authors to
take a cue from this study and cite it.https://doi.org/10.3390/toxins14080518
R. We tried to improve the organization of information (L. 148 – 154) using the study
cited as a reference. Thanks!
4. My biggest concern is for the figure 3. how come in all the graphs the standard
deviation is so high? is there an explanation for this result?
R. Figures 3 (a), (b), and (c) are the results of the Exploratory Behavior of Larvae that
were analyzed showing the immobility of the larvae, average speed, and distance traveled
by each larva individually, and indeed some larvae behaved differently from the majority
of the group, and how the graphs were made based on the averages of each group, these
outlier larvae increased the standard deviation of the groups, which is why we used 7
replicates with 15 larvae for each experimental group, trying to reduce the standard
deviation.
5. Moreover,I have doubts about these graphs, group F for example has the highest %
of motionless animals compared to the control, but they also have high speed and
good distance traveled if we compare them with other groups, please explain.
R. This difference between results may be related to how zebrafish larvae can express
varied behaviors (anxiety versus anxiolytic) (Stewart et al., 2011). Qian et al. (2019),
observed anxiety-like behaviors such as high speed (40 μg/L). In contrast to anxietyrelated
responses to fipronil, the study by Da Costa Chaulet et al. (2019) demonstrated
that exposure to fipronil (9 and 18 μg/L) induced anxiolytic behavior in zebrafish. In their
study, it was reported that fipronil did not affect the total distance traveled, however,
animals exposed to fipronil spent longer at the top of the tank, suggesting anxiolytic
properties of fipronil. Park et al. (2020) observed that fipronil induced degeneration of
motor neurons in the spinal cord and suppressed the growth of motor neuron axons in
zebrafish, partly explaining any behavioral or locomotor deficits in the animals.
Referencias: Stewart, A., Maximino, C., De Brito, T.M., Herculano, A.M., Gouveia, A.,
Morato, S., Cachat, J.M., Gaikwad, S., Elegante, M.F., Hart, P.C. Neurophenotyping of
Adult Zebrafish Using the Light/Dark Box Paradigm, Zebrafish Neurobehavioral
Protocols. Springer, 2011. pp. 157–167.
Qian, Y., Ji, C., Yue, S., Zhao, M. Exposure of low-dose fipronil enantioselectively
induced anxiety-like behavior associated with DNA methylation changes in embryonic
and larval zebrafish. Environ. Pollut, 2019. 249, 362–371.
da Costa Chaulet, F., de Alcantara Barcellos, H.H., Fior, D., Pompermaier, A., Koakoski,
G., da Rosa, J.G.S., Fagundes, M., Barcellos, L.J.G. Glyphosate- and fipronilbased
agrochemicals and their mixtures change zebrafish behavior. Arch. Environ. Contam.
Toxicol, 2019. 77 (3), 443–451.
Park, H., Lee, J.Y., Park, S., Song, G., Lim, W. Developmental toxicity of fipronil in the
early development of zebrafish (Danio rerio) larvae: disrupted vascular formation with
angiogenic failure and inhibited neurogenesis. J. Hazard. Mater, 2020. 385, 121531.
6. Still in figure 3 (a) how the F+S group is not significant compared to the control?
from the graph it seems quite different.Please explain this data
R. This group presented a value of p = 0.13084 (data from statistical analysis), and we
considered p < 0.05, to be significantly different. However, we did observe that this group
showed significant differences in figures (b) and (c), which indicates a synergistic effect
of these substances on behavior (Figure (c)). Thanks for the suggestions!

Reviewer 3 Report

Comments and Suggestions for Authors

Authors aimed to analyze the effect of different pesticides, given separately or together, on the behavior of larval zebrafish. I am not impressed with this study, there are methodological errors which makes data not reliable, and paper should not be published.

here are methodological errors. In the current version, experiments could not be replicated by other labs (poor description of methods). There is lack of information what are the differences between all chemicals, weak keywords, what does it mean “natural body waters”?

Line 49: rather optic tectum but not thalamus

What exactly was concentration of dmso? iT is very serious concern which disqualifies this paper (it is widely known that in such high concentration DMSO itself affects behavior!!!!!)

It is unacceptable to use the same larvae for so many testes- they previous may affect the results of next!!!!!

Thigmotaxis description is written in unclear way. 

What was the reason to analyze 3 min only exploratory behavior? It is a cardinal mistake for larvae.

All together I do not recommend publication of this paper because data may be misleading. 

Comments on the Quality of English Language

its ok

Author Response

We are grateful for your careful review and critique of the manuscript. Your concerns are addressed below.

There is lack of information what are the differences between all chemicals, weak keywords, what does it mean “natural body waters”?

Response: In the introduction when describing pesticides, we focused on known neurotoxic effects to justify the use of behavioral tests, and in the discussion, we better explain their modes of action. We added brief descriptions about the compounds in the introduction. Thanks for the suggestion.

We try to improve the keywords, according to your suggestion.

We meant ‘natural body of water’, i. e., any spring, stream, pond, lake, or wetland that was historically present in a natural state but may have been physically altered over time. We changed this term to "freshwater" for brevity and clarity.

Line 49: rather optic tectum but not thalamu

Response: This term was taken from the article: Wu, C.H.; Lu, C.W.; Hsu, T.H.; Wu, W.J.; Wang, S. E. Fipronil neurotoxicity affects zebrafish sensory and motor systems. Pestic Biochem Physiol 2021, 177, 104896. And as we are using this article as a reference, we chose to keep this term. We added optic tectum in parentheses. Thanks!

What exactly was concentration of DMSO? iT is very serious concern which disqualifies this paper (it is widely known that in such high concentration, DMSO itself affects behavior!!!!!)

Response: The DMSO concentration as described in the methodology L. 108 “The final nominal concentration of DMSO was less than 1% (v/v)”. This method was used because zebrafish are tolerant to DMSO (≤1%) (Zhang et al., 2023). Therefore, there is no interference in behavioral tests based on your results.

Reference: Zhang, Z.; Qiu, T.; Zhou, J.; Gong, X.; Yang, K.; Zhang, X.; Ji, Y. Toxic effects of sirolimus and everolimus on the development and behavior of zebrafish embryos. Biomed  Pharmacothep 2023, 166, 115397.

It is unacceptable to use the same larvae for so many testes- they previous may affect the results of next!!!!!

Response: All tests were previously approved by the ethics committee, which prioritizes reducing the number of animals used in tests. The tests carried out without major chemical or physical interventions and without causing excessive stress to the animals since most of the analyses are done later using photos and videos.

An interval of 15 minutes was also used between tests for the larvae to acclimatize. Cadena et al. (2020a), the article we used as a reference to carry out this multibehavioral analysis, used 10 min of acclimatization between tests. Here, a video system was used, as described in text L. 143-144.

Reference: Cadena, P.G.; Cadena, M.R.S.; Sarmah, S.; Marrs, J.A. Folic acid reduces the ethanol-induced morphological and behavioral defects in embryonic and larval zebrafish (Danio rerio) as a model for fetal alcohol spectrum disorder (FASD). Reprod Toxicol 2020, 96, 249-257, doi:10.1016/j.reprotox.2020.07.013.

Thigmotaxis description is written in unclear way.

Response: The articles used as references are cited in the text, and the graphic of this test has an illustrative figure to facilitate understanding of the test. But we also rewrote some parts of the text to improve the test description.

What was the reason to analyze 3 min only exploratory behavior? It is a cardinal mistake for larvae.

Response: This test was also based on the methodology of Pérez-Escudero et al. (2014) this reference and more information were added to the article.

After acclimatization for 15 min, the fish were placed on an LED board where they were observed and recording was carried out for subsequent analyses for 5 min, as per Altenhofen et al. (2017). However, data from the first and last minutes were discarded, because it was observed that in the first minute the larvae did not show swimming behaviors due to exposure to artificial light from the LED board and in the last minute the animals also reduced the display of swimming behaviors, as was also observed by Pérez-Escudero et al. (2014). And we emphasize that this reduction in analysis time did not affect the assay results.

Reviewer 4 Report

Comments and Suggestions for Authors

I have only a few demands, explain the figures, check how to cite references in the text and check references that ae missing assessing date

Author Response

We are grateful for your careful review and critique of the manuscript. Your concerns are addressed below.

Table 1. explain shortnames.

Response: The explanation of "dpf" is in L.74 "days post fertilization". But we also added the table. Thanks for the suggestion!

122. DO?

Response: "DO" is Dissolved Oxygen, we added the explanation to the text.

Figure 1. SD?

Response: "SD" is Standard Deviation, we added the explanation to the figure captions.

164-165. Triplicates?

Response: The tests were carried out in triplicates and the results were shown in mean and standard deviation.

288-292. SD? triplicates?

Response: We made the changes according to your suggestions.

362, 373, 398. Authors of the article.

Response: We made changes to the citations in the text according to your suggestions.

559, 561, 564. Accessed on.

Response: We have added access dates. Thanks.

Reviewer 5 Report

Comments and Suggestions for Authors

The article is original and very relevant for the field. The authors studied the impact of  3 pesticides, both individually and in mixtures, on zebrafish larvae using a multi-behavioral approach. This approach proved to be a sensitive and comprehensive tool for analyzing the systemic effects of one or more substances in zebrafish larvae.

The results showed that, sulfentrazone caused the greatest number of behavioral changes in exposed larvae. Fipronil, caused immobile behavior. Carbendazim had less pronounced effects, but still affected the animals' escape and grouping behavior. When the authors analyzed pesticide mixtures, they observed synergistic responses that affected speed, distance covered, and opto motor response.

The methology of the study is modern and very complex.

The conclusions are consistent with the evidence and arguments presented. The results of the study offer further evidence that continued research in the area of of the toxicity of pesticide mixtures and the sensitivity of behavioral tests, which can be used as initial indicators of environmental stress.

The references are appropriate, including some relevant authors experience in the field.

I recommend some corrections.

1.     The manuscript should be thoroughly evaluated by a Native English speaker.

2.     In Introduction you should mention briefly for what are used these pesticides and when are used together

3.     When you cite some authors in text, don t mention all the authors of the article, just the first one et al.

4.     References should be written according to Instructions for authors. Ref 32-43 mention accessed on.(when?)

You may see also

Ilie, O.-D.; Duta, R.; Balmus, I.-M.; Savuca, A.; Petrovici, A.; Nita, I.-B.; Antoci, L.-M.; Jijie, R.; Mihai, C.-T.; Ciobica, A.; et al. Assessing the Neurotoxicity of a Sub-Optimal Dose of Rotenone in Zebrafish (Danio rerio) and the Possible Neuroactive Potential of Valproic Acid, Combination of Levodopa and Carbidopa, and Lactic Acid Bacteria Strains. Antioxidants 202211, 2040. https://doi.org/10.3390/antiox11102040

Comments on the Quality of English Language

  The manuscript should be thoroughly evaluated by a Native English speaker.

Author Response

We are grateful for your careful review and critique of the manuscript. Your concerns are addressed below.

The manuscript should be thoroughly evaluated by a Native English speaker.

Response: The manuscript was revised. Due to the extensive changes, these were not highlighted for clarity. Thanks.

In Introduction you should mention briefly for what are used these pesticides and when are used together

Response: We added the introduction when pesticides are used together according to your suggestion.

When you cite some authors in text, don t mention all the authors of the article, just the first one et al.

Response: We made changes to the references according to your suggestion.

References should be written according to Instructions for authors. Ref 32-43 mention accessed on.(when?)

Response: We added access dates according to your suggestion. Thanks for the suggestion.

Round 2

Reviewer 1 Report

Comments and Suggestions for Authors

The corrections have been made according to the reviewers' comments and the article would be published in the present form

Author Response

Thank you for the positive feedback on our manuscript.

Reviewer 2 Report

Comments and Suggestions for Authors

The authors dispelled my misgivings for the most part. However, the bibliography has not been updated . 

Lines 149-155: references number 38-39 in the manuscript refer to other publications other than the answer provided by the authors. Please explain

Author Response

We are grateful for your careful review and critique of the manuscript. Your concerns are addressed below.

Lines 149-155: references number 38-39 in the manuscript refer to other publications other than the answer provided by the authors. Please explain.

Response: Reference 38 corresponds to Cadena et al., 2020a, and reference 39 Cadena et al., 2020b. Reference 38 was used to refer to behavioral tests with zebrafish larvae and reference 39 was used as a reference to tests with mixtures of chemicals using a simplex centroid design. Both papers were published by the same first author using zebrafish. Thanks.

“38. Cadena, P.G.; Cadena, M.R.S.; Sarmah, S.; Marrs, J.A. Folic acid reduces the ethanol-induced morphological and behavioral defects in embryonic and larval zebrafish (Danio rerio) as a model for fetal alcohol spectrum disorder (FASD). Reprod Toxicol 2020, 96, 249-257, doi:10.1016/j.reprotox.2020.07.013.

39. Cadena, P.G.; Sales Cadena, M.R.; Sarmah, S.; Marrs, J.A. Protective effects of quercetin, polydatin, and folic acid and their mixtures in a zebrafish (Danio rerio) fetal alcohol spectrum disorder model. Neurotoxicol Teratol 2020, 82, 106928, doi:10.1016/j.ntt.2020.106928.”

Reviewer 3 Report

Comments and Suggestions for Authors i ve rejected this paper initially, and I ve not changed my mind. For me there are mistakes, which are cardinal. Authors did not convince me at all.   DMSO mentioned: https://pubmed.ncbi.nlm.nih.gov/37406269/ this paper is now recommended! above 0.55% DMSO substantially affects behavior.
  Influence of Methylene Blue or Dimethyl Sulfoxide on Larval Zebrafish Development and Behavior - PubMed pubmed.ncbi.nlm.nih.gov The use of larval zebrafish developmental testing and assessment, specifically larval zebrafish locomotor activity, has been recognized as a higher throughput testing strategy to identify developmentally toxic and neurotoxic chemicals. There are, however, no standardized protocols for this type of a …

  The fact that study was approved my EC does not change the fact, that usually members (still) do not know much about zebrafish. The community is still very small, and I will not contribute for publishing, in my opinion, methodological errors.       Authors wrote that used 840 larvae, 15 times replicated. They could easily use each bach (but bigger) for one experiments and this is recomended among "fish people". They do not know the rules.    I do not except this paper. I am sorry.   I will not sign to accept this paper.

Author Response

We are grateful for your careful review and critique of the manuscript. Your concerns are addressed below.

DMSO mentioned: https://pubmed.ncbi.nlm.nih.gov/37406269/ this paper is now recommended! above 0.55% DMSO substantially affects behavior.

Response: All experimental groups were exposed to the same DMSO concentration and significant differences (p < 0.05) were found between groups in behavioral analyses. The final DMSO concentration in the embryo medium was less than 0.01% (v/v), because we prepared a pesticide stock solution with 1% of DMSO and then diluted these solutions in the embryo medium to obtain the pesticide concentrations used in the study. The final concentration of DMSO is 50 times lower than that reported in the cited reference (0.5% v/v) by the reviewer. Therefore, the reference indicated should not disqualify our work. We understand that this detail was not described in the initial version of the paper, but we have now modified the text according to your criticisms.

Authors wrote that used 840 larvae, 15 times replicated. They could easily use each bach (but bigger) for one experiments and this is recomended among "fish people".

Response: We rewrote the text for better understanding, as “105 larvae (15 x 7 authentic replicates, 8 experimental groups ≈ 840 larvae) were used for each experimental group. All larvae went through the same sequence of behavioral tests, thigmotaxis, sensitivity to touch, optomotor response, larval exploratory behavior, and bouncing balls, respectively", different from "used 840 larvae, 15 times replicated" as understood by the reviewer.

Concerning the animal welfare committee, we understand your concerns, but it is necessary to comply with the committee, whether they are "fish people" or not. Thanks.

Reviewer 5 Report

Comments and Suggestions for Authors

The authors have made all the corrections suggested. I recommend the acceptance of the article in present revised form

Author Response

(The authors gave the same response as above.)
